# A Low-Cost Electrochemical Method for the Determination of Sulfadiazine in Aquaculture Wastewater

**DOI:** 10.3390/ijerph192416945

**Published:** 2022-12-16

**Authors:** Yang Liu, Jianlei Chen, Haiyan Hu, Keming Qu, Zhengguo Cui

**Affiliations:** 1Faculty of Fisheries, Zhejiang Ocean University, Zhoushan 316022, China; 2Key Laboratory of Sustainable Development of Marine Fisheries, Ministry of Agriculture and Rural Affairs, Yellow Sea Fisheries Research Institute, Chinese Academy of Fishery Sciences, Qingdao 266071, China; 3College of Marine Science and Technology, Zhejiang Ocean University, Zhoushan 316022, China

**Keywords:** antibiotics, sulfadiazine, electrochemical sensors, aquaculture, wastewater

## Abstract

As the concept of green development spreads worldwide, environmental protection awareness for production and life has been continuously strengthened. Antibiotic residues in aquaculture wastewaters aggravate environmental pollution and threaten human health. Therefore, the detection of residual antibiotics in wastewater is crucial. In this paper, a new, simple, and low-cost method based on the glassy carbon electrode electrochemical sensor for the detection of sulfadiazine in aquaculture wastewater was developed without using complex materials to modify the electrode surface, to detect sulfadiazine which electrochemically oxidizes directly. The electrochemical performance of the sensor was studied and optimized with differential pulse voltammetry and cyclic voltammetry in the three-electrode system. The optimal electrolyte was acetic acid-sodium acetate buffer, and the optimal pH was 4.0. Finally, based on the optimized conditions, the newly established method showed satisfactory results for detecting sulfadiazine in aquaculture wastewater. The concentration of sulfadiazine and the peak current intensity showed a linear relationship in the range of 20 to 300 μmol/L, and the limit of detection was 6.14 μmol/L, the recovery rate of standard addition was 87–95%, with satisfactory reproducibility and low interference.

## 1. Introduction

Antibiotics refer to some secondary metabolites with anti-pathogen functions produced by microorganisms or other animals during the life process. With the development of modern society and the progress of the industrial field, as well as people’s production and life, the use of antibiotics is also increasing. The abuse of antibiotics in humans and animals can lead to the drug resistance of microorganisms [1], affects biological communities [2], and destroys the ecosystems that the antibiotics enter, thereby posing a serious threat to human health [3]. At present, it is also difficult to directly treat and degrade these pollutants by natural purification alone, which leads to the aggravation of environmental pollution. Further, sewage treatment technology is complicated, and the cost is high, therefore various antibiotics in the wastewater cannot be completely removed [4,5,6,7]. Therefore, the detection of residue antibiotics in wastewater is of great significance for understanding the efficiency of the water treatment process and protecting human health.

Sulfadiazine (SDZ) is one of the sulfonamide antibiotics (SAs), which is widely used in human medical treatment and aquaculture due to its strong bactericidal properties and convenience [8]. SDZ is one of the most widely used antibiotics in the world and is frequently detected as a trace contaminant in wastewater around the world [9,10]. SDZ has strong biochemical stability due to the inherent N, S heterocyclic structure [11], and it is not easily absorbed in animals, so a large part of it is easily excreted as metabolites, thereby entering the environment [12]. Trace SAs residues in water have ecotoxicological effects on aquatic organisms, destroy the ecological environment, and transfer to humans through bioaccumulation, posing a threat to human health [13,14]. The detection of residual antibiotics in wastewater has been a most concerning issue at this time. There are various methods for the detection of SDZ at present, including high-performance liquid chromatography (HPLC) [15,16], liquid chromatography-mass spectrometry [17], spectrophotometry [18], and capillary electrophoresis [19,20]. Although these methods are reliable and accurate, they have the disadvantage of being expensive instruments with a long analysis time. For example, the sample pretreatment process for HPLC analysis is cumbersome and time-consuming, besides, the equipment and its maintenance costs are expensive, which increases the detection cost; immunoassays and capillary electrophoresis methods may have poor repeatability, and the specificity and sensitivity also need to be increased.

As an important analytical technique, the electrochemical technique converts a chemical quantity of the analyte into an electrical quantity for detection based on the electrochemical properties of the analyte [21]. The electrochemical sensor can be prepared easily and at a low cost, which has the characteristics of fast analysis speed, high specificity, and good sensitivity for detecting the target analyte. Most importantly, it can realize some online monitoring that cannot be realized by traditional methods. Therefore, the research into detection and analysis by constructing electrochemical sensors has gradually increased in recent years. Many methods have been reported on electrochemical sensors to detect SDZ. Sun et al. (2019) [22] modified an electrode with an organic framework, and then used SDZ and acetaminophen as templates to polymerize pyrrole on the modified electrode, and formulated electrochemical sensing for the simultaneous determination of SDZ and acetaminophen. Kokulnathan et al. (2021) [23] used the acoustic chemical synthesis of strontium tungstate nanosheets to modify screen-printed carbon electrodes, and an electrochemical sensor for detecting sulfadiazine in environmental samples was constructed. Ding et al. (2021) [24]. prepared two types of quantum dots attached to quantum dots using 3-aminopropyltriethoxysilane as a functional monomer, and sulfadimethoxine and SDZ as template molecules, molecularly imprinted as a silica layer on the surface for SDZ detection in seawater and shrimp samples.

The above-mentioned electrochemical detection methods are used to modify the electrode to amplify the electrochemical signal and then directly or indirectly detect SDZ in the sample. Although these methods amplified the electrochemical signal of the electrochemical sensor, they would have a higher cost of modification, complicated configurations and can easily fall off. It is simple, time-saving, and low-cost to manufacture sensors without using complex materials, and it is conducive to the implantation of this method in rapid routine analysis. In order to minimize the cost and reduce the detection time, the unmodified electrochemical sensor was characterized with a differential pulse voltammetry curve and cyclic voltammetry curve. The background solution and pH of the measured SDZ were optimized. Furthermore, considering the complexity of aquaculture wastewater, artificial seawater was regarded as a significant influence factor for optimization. We wanted to reduce the detection costs as much as possible and make the preparation and detection process of electrochemical sensors simpler. Finally, the newly proposed method based on an electrochemical sensor was applied to detect the SDZ in aquaculture wastewater.

## 2. Experimental Section

### 2.1. Reagents and Instruments

Sulfadiazine (SDZ) was purchased from Aladdin Biochemical Technology Co., Ltd. (Shanghai, China). Acetonitrile was purchased from Merck. Acetic acid was purchased from Tianjin Kemeiou Chemical Reagent Co., Ltd. (Tianjin, China). Sodium dihydrogen phosphate (NaH_2_PO_4_), disodium hydrogen phosphate (Na_2_HPO_4_), potassium chloride (KCl), sulfuric acid (H_2_SO_4_), sodium hydroxide (NaOH), sodium chloride (NaCl), calcium chloride (CaCl_2_), magnesium chloride (MgCl_2_), sodium sulfate (Na_2_SO_4_), hydrochloric acid (HCl), sodium acetate, strontium chloride (SrClz 6H_2_O), potassium bromide (KBr), sodium bicarbonate(NaHCO_3_), sodium fluoride (NaF), boric acid (H_3_BO_3_), potassium ferricyanide (K_3_Fe(CN)_6_), and ferrous iron potassium cyanide (K_4_Fe(CN)_6_) was obtained from Sinopharm Chemical Reagent Co., Ltd. (Shanghai, China). All reagents were of analytical grade, and the water was ultrapure (18 MΩ cm, 25 °C).

The electrochemical workstation was a CHI660D (CH Instrument Company, Shanghai, China) with a three-electrode system, a glassy carbon electrode (GCE, 3 mm diameter) as the working electrode, a Ag/AgCl (saturated KCl) reference electrode, a platinum wire electrode as the auxiliary electrode; scanning electron microscopy (SEM) images were provided using a regulus 8100 (Hitachi, Nako, Japan).

### 2.2. Pretreatment of Electrodes

GCEs were polished to a mirror surface using different diameters of Al_2_O_3_ (0.5, 0.3, and 0.05 μm) on the chamois in turn, and the polishing leather was purchased from Jingchong Electronic Technology Development Co., Ltd. (Shanghai, China). The polishing time was 30 s each time, rinsed with ultrapure water, then ultrasonically washed in ethanol and ultrapure water for about 2–3 min. After the electrode was dried with nitrogen, it was placed in 5 mmoL/L of Fe(CN)_6_^3−^/Fe(CN)_6_^4−^ for cyclic voltammetry (CV) measurement to ensure that the potential difference between oxidation peak and reduction peak was less than 0.1 V.

### 2.3. Preparation of the Standard Solution

The SDZ standard stock solution (5 mmoL/L) was prepared with acetonitrile. The different concentrations of the standard solution (20–300 μmol/L) were diluted with acetic acid-sodium acetate buffer solution (ABS, pH 4). The ABS buffer solution was prepared by dissolving 18 g of sodium acetate and 9.8 mL of acetic acid (12.063 mol/L) in 1000 mL of water. Furthermore, the pH of different ABS buffer solutions was obtained by adjusting with HCl (2 mol/L and 6 mol/L) and NaOH (2 mol/L and 10 mol/L).

### 2.4. Electrochemical Determination of SDZ

Aquaculture wastewater samples were collected from Haiyang Yellow Sea Aquatic Product Co. Ltd. and then analyzed using the proposed electrochemical method for SDZ determination. The samples were spiked using the following protocol. Firstly, take 5 mL of water sample in a glass bottle, add 100 μL of SDZ standard solution (5 μmol/L), add ABS buffer solution up to 10 mL, mix thoroughly, and adjust the pH to 4. Lastly, the working electrode was immersed to perform a quantitative analysis of SDZ by differential pulse voltammetry (DPV) under optimal conditions. DPV was used to record the current response in the presence of SDZ in the potential range of 0.6 to 1.2 V.

## 3. Results and Discussion

### 3.1. Polishing of the Electrode

The degree of electrode grinding directly affects the sensitivity of the electrodes, which affects the accuracy of the measurement result. In order to ensure stable measurements, the working electrode needed to be pre-treated by polishing to guarantee an excellent- electrical signal response which could achieve 0.1 V under cyclic voltammetry. As shown in Figure 1, the same electrode was repeatedly polished five times, and CV scanning was performed once in a redox solution. After five times of polishing, the CV curves of the electrode showed little difference, which indicated that the electrode itself was reproducible. Cyclic voltammograms were collected from −0.2 V to 0.8 V at a scan rate of 50 mV/s in the Fe(CN)_6_^3−^/Fe(CN)_6_^4−^ redox probe solution.

The surface morphologies of the polished and unpolished glassy carbon electrodes were characterized with SEM. As shown in Figure 2, the polished electrode surface was rough, which increased the electrochemically active surface area, and the unpolished electrode surface was passivated and smooth. In addition, the electrochemically active surface areas of the GCEs were calculated using the Randles-Sevcik Equation (1) [22] as follows:I_p_ = (2.69 × 10^5^) n^3/2^AD^1/2^Cv^1/2^
(1)
where n is the number of electrons produced in the reaction, equal to 1, v is the scan rate equal to 50 mV/s, A is the electroactive surface area of the electrode, D is the Fe(CN)_6_^3−^/Fe(CN)_6_^4−^ diffusion coefficient equal to 7.6 × 10^−5^ cm^2^/s, C is the Fe(CN)_6_^3−^/Fe(CN)_6_^4−^ concentration (5 mmol/L), and Ip is the peak current (A). For the GCE in this study, the electroactive surface area was found to be 0.12 cm^2^.

### 3.2. Comparison of Electrolyte Solutions

Considering the complexity of aquaculture wastewater components, the background solution for SDZ detection was optimized. The experiments were performed in NaCl (3.8%) solution, ABS buffer solution (pH 4), artificial seawater (Mocledon’s artificial seawater formula at pH 4 [25,26,27,28]), and 0.2 mol/L of phosphate buffer solution containing SDZ (200 μmol/L) using DPV measurements. The NaCl (3.8%) solution was prepared by dissolving 3.8 g NaCl solid in 100 mL of water. The ABS buffer solution (pH 4) and artificial seawater were prepared as mentioned above. In addition, the pH of all used buffer solutions was adjusted with HCl and NaOH. As shown in Figure 3, SDZ had the highest current response intensity in the ABS buffer solution, so the ABS buffer solution was selected as the optimal background electrolyte solution.

### 3.3. Comparison of Electrolyte pH Affecting Current Strength

To explore the effect of pH on detecting SDZ, the various ABS buffer solutions (pH range from 2 to 8) were investigated. As the pH increased from 2 to 8, the oxidation peak position of SDZ shifted to the left, and the oxidation peak current of SDZ was the largest when the pH was 4 (Figure 4). This was because the SDZ can be electrochemically oxidized at the –NH_2_ group by a two-electron and two-proton transfer process, as shown in Appendix A. Further, it reveals that the oxidation of SDZ occurs at 0.9–1.0 V, which is consistent with previous reports [29,30,31,32,33]. The increased hydrogen ion concentration promotes the oxidation process at pH 4. Therefore, the ABS buffer solution with a pH of 4 was selected as the best-supporting electrolyte for the experiment.

It was found that there was a linear relationship between the scanning rate and the CV curve containing the SDZ electrolyte, from which the relationship between E_p_ and ln v could be calculated, and the heterogeneous electron transfer rate constant (k_s_) could be calculated according to the Laviron Equation (2) [34]:E_p_ = E^0^ + RT/(anF) [ln [(RTk_s_)/(anF)] − ln v](2)
where E_p_ is anode peak potential, E^0^ is formal potential, R is the gas constant (8.314 J/mol·K), T is the temperature (298.15 K), α is the electron transfer coefficient and calculated based on the linear relationship between E_p_ and ln v, n is the charge transfer number, F is the Faraday constant, Ks is the electron transfer rate constant of the surface process, and υ is the scanning rate. Lastly, K_s_ was calculated as 1.13 s^−1^.

### 3.4. Analytical Performance of the Sensor

To evaluate the analytical performance of the constructed sensor, 10 mL of SDZ standard solution was detected by DPV in ABS buffer solution at pH 4. As shown in Figure 5, the concentration of SDZ and the generated peak current have a good linear relationship in the range of 20–300 μmol/L, the linear regression equation could be described as Ip (μA) = 0.0269 c (μmol/L) + 1.2, the correlation coefficient (R^2^) was 0.994, and the limit of detection (LOD) was 6.14 μmol/L which was calculated by the following equations: LOD = 3 σ/b (where σ is the standard deviation obtained from five times measurements of the blank signal and b is the analytical sensitivity represented by the slope of the calibration plot).

The performance of the proposed method was compared with previously reported SDZ sensors in the literature, as shown in Appendix A. Compared with other unmodified electrodes, this sensor had a satisfactory electrocatalytic performance.

### 3.5. Sensor Reproducibility and Interference Immunity

To study the stability of the electrode, after the electrode was used, it was ground again to form a peak potential difference in the cyclic voltammogram of 0.1 V, and the electrode could be used again. The treated glassy carbon electrode was subjected to DPV measurements in 50 μmol/L SDZ solution under optimized conditions which were repeated 5 times.As shown in Figure 6. The relative standard deviation (RSD) of the current response signal was 2.23% which showed that the electrode had satisfactory reproducibility.

In this experiment, the interference effect of inorganic ions that may exist in the actual sample on the determination of SDZ was investigated. Under optimal conditions, 25 mmol/L of KCl, CaCl_2_, MgCl_2_, and Na_2_SO_4_ were added to the SDZ (50 μmol/L) standard solution. The enhancement of a group of current signals with interfering substances is shown in Figure 7. and the electrical signal changes by 3.03%, within the tolerance range of 5%. It was indicated that 250 times Cl^−^, 100 times Na^2+^ and 50 times K^+^, Mg^2+^, Ca^2+^, and SO_4_^2−^ were not able to interfere with the SDZ measurement signal, which proved that the electrochemical sensor had-good anti-interference abilities.

### 3.6. Practical Sample Application

In order to evaluate the practical application of the proposed method, it was used for the determination of SDZ in aquaculture wastewater. The measurement results are shown in Table 1. The recovery rate of the standard addition was between 87% and 95%, indicating that the constructed sensor had high accuracy and could be used for the analysis of SDZ in actual samples.

## 4. Conclusions

In conclusion, a low-cost method based on an electrochemical sensor for the detection of SDZ was constructed in this paper. The polishing of the electrode was a crucial part of this experiment, and the repeatability of the tested electrode was discussed with the same electrode after polishing five times. The type and pH of the electrolyte were optimized to achieve a sensor with high performance, and the ABS buffer solution with pH 4 was used as the optimal electrolyte. The sensor had a satisfactory linear response to SDZ in the concentration range of 20–300 μmol/L and was successfully applied to analyze actual aquaculture wastewater. The proposed method has satisfactory sensitivity, reproducibility, and a simple operation, which provides a new tool idea for the detection of sulfonamide antibiotics in aquaculture wastewater.

## Figures and Tables

**Figure 1 ijerph-19-16945-f001:**
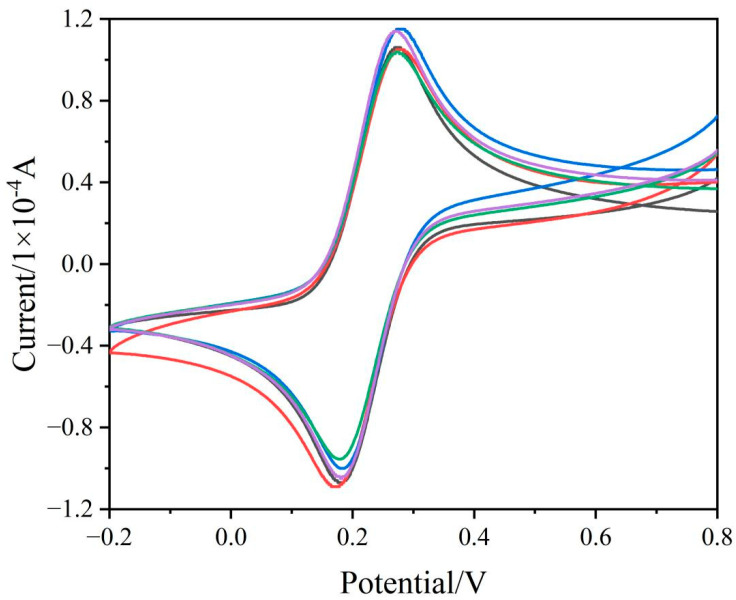
Cyclic voltammetry (CV) curve of the same electrode in 0.1 mol/L KCl containing 5 mmol/L of Fe(CN)_6_^3−^/Fe(CN)_6_^4−^ after polishing (*n* = 5).

**Figure 2 ijerph-19-16945-f002:**
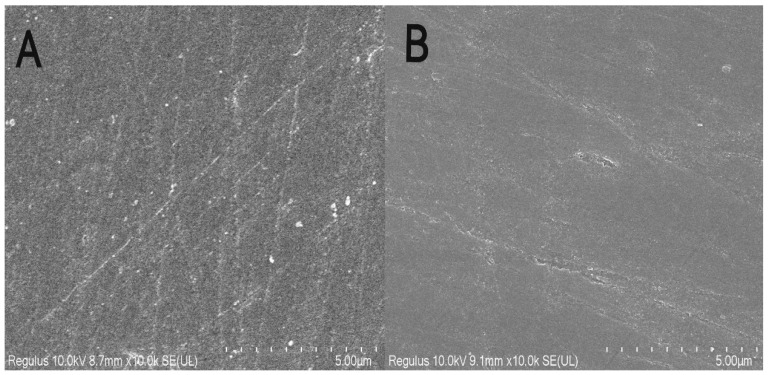
SEM image of glassy carbon electrode. (**A**), after polishing the electrode; (**B**), before polishing the electrode.

**Figure 3 ijerph-19-16945-f003:**
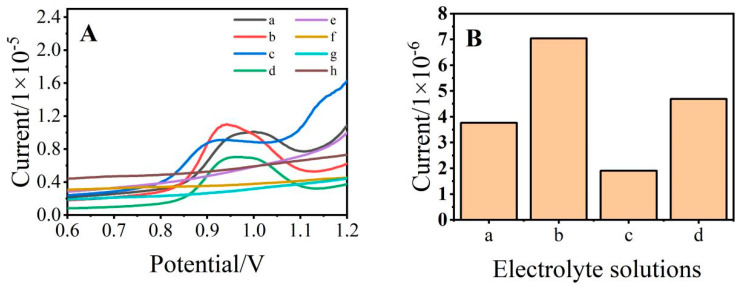
(**A**) Differential pulse diagram of sulfadiazine (SDZ) (**a**–**d**) and without SDZ (**e**–**h**) detection in different electrolyte solutions; (**B**) Current response of SDZ detection in different electrolyte solutions; ((**a**), NaCl (3.8% pH 4); (**b**), acetic acid-sodium acetate buffer solution (pH 4); (**c**), artificial seawater (pH 4); (**d**), phosphate buffer solution (pH 4); and (**e**–**h**) refers to (**a**–**d**) electrolyte solution without SDZ).

**Figure 4 ijerph-19-16945-f004:**
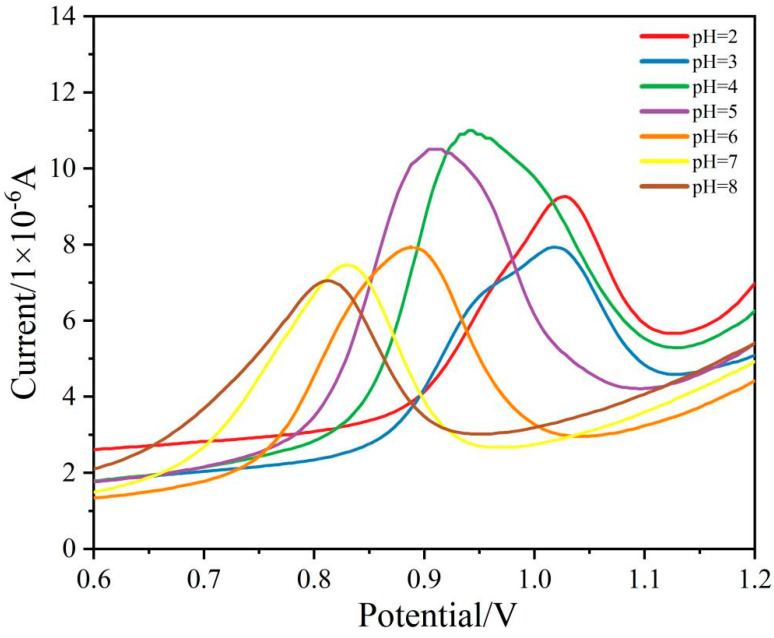
Differential pulse diagram of SDZ detection under different pH conditions.

**Figure 5 ijerph-19-16945-f005:**
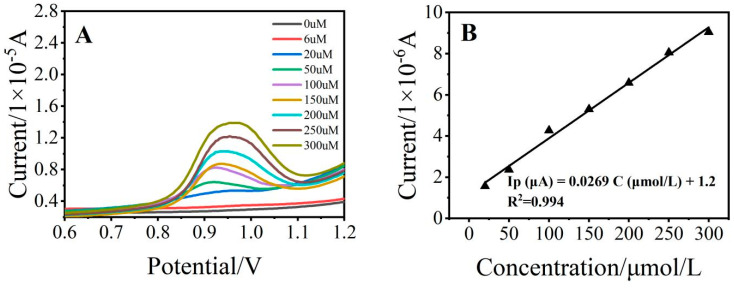
(**A**) Differential pulse diagram of the glassy carbon electrode for different concentrations of SDZ standard solution; and (**B**) Linear relationship between different concentrations of SDZ and the oxidation current.

**Figure 6 ijerph-19-16945-f006:**
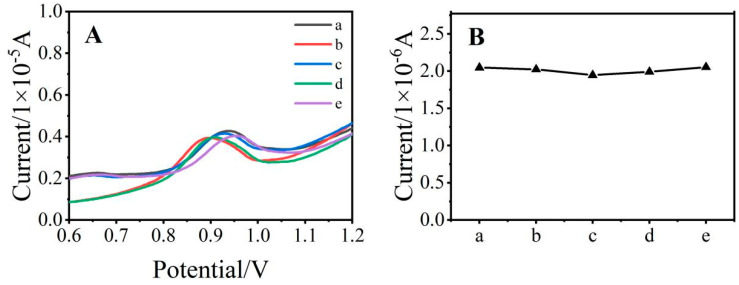
Differential pulse diagram (**A**) and current response (**B**) of the same electrode for SDZ detection five times (50 μmol/L, (**a**–**e**) are first to fifth).

**Figure 7 ijerph-19-16945-f007:**
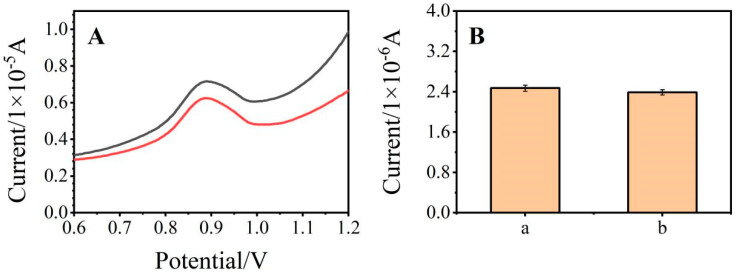
Differential pulse curve (**A**) and current response (**B**) of acetic acid-sodium acetate buffer solution containing 50 μmol/L of SDZ detected by the same electrode under the same conditions ((**a**), interfering substances added; and (**b**), without interfering substances added).

**Table 1 ijerph-19-16945-t001:** Determination of sulfadiazine in an aquaculture wastewater sample.

Samples	Detection (μmol/L)	Spiked Level (μmol/L)	Found Level (μmol/L)	Recovery %
Aquaculture wastewater	nd	50.00	43.75	87.50
	nd	100.00	92.75	92.75
	nd	150.00	135.76	90.51

Note: “nd” means that the sample does not contain the detected substance.

## Data Availability

Not applicable.

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
