# Peer review of "A Low-Cost Electrochemical Method for the Determination of Sulfadiazine in Aquaculture Wastewater"

_ijerph, 2022, doi:10.3390/ijerph192416945_

Round 1

Reviewer 1 Report

Presented method is well-optimized, practical and reliable, using a simple and commonly available glassy carbon electrode. The overall quality and scientific impact of the manuscript is, however, below average. Writing style, grammar, typography and several other parts of the manuscript needs to be improved, specific comments are in the attached PDF. 

Author Response

Response to Reviewer 1 Comments

Dear Prof. Aiyana Zhang and Reviewer,

We greatly acknowledge the editor and reviewer for their constructive comments and advice. Thank you for your patience and these valuable comments. It is our honor to get your help to improve our manuscript entitled “A low-cost electrochemical method for the determination of sulfadiazine in aquaculture wastewater” with the manuscript number ijerph-2058366.

We have revised the manuscript according to every single comment which made by the editor and reviewers. We check the editor and reviewers’ comments repeatedly and closely to ensure each comment is addressed. The following is our reply to the comments in red color. Moreover, the changes have been highlighted in red color in a marked copy of the revised manuscript (in submitted documents).

Response to Reviewer #1:

General comments

Presented method is well-optimized, practical and reliable, using a simple and commonly available glassy carbon electrode. The overall quality and scientific impact of the manuscript is, however, below average. Writing style, grammar, typography and several other parts of the manuscript needs to be improved, specific comments are in the attached PDF

Response: Thank you for your carefully reading and valuable comments. We have revised carefully according to the comments and try our best to improve the paper. 

Specific comments

  1. Line 14: add the specific type of the used sensor e.g. glassy carbon electrode

Response: Thank you for your comment. Correct according to comments as follows now. (Line 13-17)

“In this paper, a new method based on the glassy carbon electrode electrochemical sensor for the detection of sulfadiazine in aquaculture wastewater was developed without using complex materials to modify the electrode surface, and detecting the sulfadiazine with electrochemically oxidizes directly, which was a simple operation and low cost.”

  1. Line 21: write "low interference"

Response: Thank you for your comment. Correct according to comments. (Line 24)

  1. Line 28: no need to write "abuse use", write "abuse"

Response: Thank you for your comment. Correct according to comments. (Line 31)

  1. Line 45: rewrite the sentence, use verb "transfer" instead of flow

Response: Thank you for your comment. Correct according to comments. (Line 49)

  1. Line 49: "mass spectrometry"

Response: Thank you for your comment. Correct according to comments. (Line 53)

  1. Line 49:no need to capitalize S

Response: Thank you for your comment. Correct according to comments. (Line 53)

  1. Line 47:please read IUPAC dictionary for the difference between a method and a technique

Response: Thank you for your valuable comment. We revised the sentence as follows now. (Lines 60-62)

“As an important analytical technique, the electrochemical technique converts the chemical quantity of the analyte into an electrical quantity for detection based on the electrochemical properties of the analyte [21].”

  1. Line 97: remove the star symbol

Response: Thank you for your comment. We have removed the star symbol. (Line 107)

  1. Line 98: add diameter

Response: Thank you for your comment. We have added diameter as follows now. (Lines 109-110)

“glassy carbon electrode (GCE, 3 mm diameter) as working electrode”

  1. Line 99: electrode

Response: Thank you for your comment. Correct according to comments. (Line 110)

  1. Line 99: potential of Ag/AgCl electrode depends on the concentration of Cl- ions, please add the concentration and type of ions. e.g. 3M, 1M or saturated KCl

Response: Thank you for your comment. It was saturated KCl. We have added the detail information in Line 110.

“Ag/AgCl (saturated KCl) reference electrode”

  1. Line 103: write "polishing leather", add the manufacturer info

Response: Thank you for your comment. We have written "polishing leather" and added the manufacturer info as follows now. (Lines 115-117)

“GCE was polished to a mirror surface with different diameters of Al2O3 (0.5, 0.3, and 0.05 μm) on the chamois in turn, the polishing leather was purchased from Jingchong Electronic Technology Development Co., Ltd (Shanghai, China).”

  1. Line 110: rewrite the sentence more clearly, is the solution 5mmol/L SDZ in acetonitrile? Or another concentration of SDZ in 5mmoL/L acetonitrile?

Response: Sorry for the confusion. It is 5mmol/L of the SDZ solution in acetonitrile, we have revised the sentence as follows. (Line 124-125)

“The SDZ standard stock solution (5 mmoL/L) was prepared with acetonitrile.”

  1. Line 113: add concentration

Response: Thank you for your comment. We have added the concentration is

12.063 mol/L. (Lines 127-128)

“The preparation of the ABS buffer solution was to dissolve 18 g of sodium acetate and 9.8 mL of acetic acid (12.063 mol/L) in 1000 mL of water.”

  1. Line 114: were

Response: Thank you for your comment. Correct according to comments. (Line 128)

16.Line 115: a tip for future experiments: buffering capacity of the used buffer is negligible in the pH 7-12, consider using phosphate or Britton Robinson buffer

Response: Thank you for your valuable comment. That's a great suggestion. We will consider using phosphate or Britton Robinson buffer in the future study.

  1. Line 121: use capital M (mol) instead of m (meter)

Response: Thank you for your comment. Correct according to comments. (Line 136)

  1. Line 128: "grinding" means you are making a powder out the electrode material. Use "polishing" instead

Response: Thank you for your comment. Correct according to comments. (Line 144)

  1. Line 132: pretreated

Response: Thank you for your comment. Correct according to comments. (Line 147)

  1. Line 133: ???

Response: Sorry for the confusion. We have used "polishing" instead and revised this sentence. (Line 146-149)

“In order to ensure the stability of the measurement, the working electrode must be pretreated to guarantee its excellent electrical signal response by polishing which could achieve 0.1 V under the cyclic voltammetry for further use.”

  1. Line 141: add concentration, and used electrolyte

Response: Thank you for your comment. We have added the concentration and used electrolyte in Figure 1. (Line 157-158).

“Figure 1. Cyclic voltammetry (CV) curve of the same electrode in 0.1 mol/L KCl containing 5 mmol/L of Fe(CN)63-/Fe(CN)64- after polishing (n=5).”

  1. Line 145: no need to use optimized twice in one sentence

Response: Thank you for your comment. we have revised the sentence as follows. (Line 177-178).

“Considering the complexity of aquaculture wastewater components, the background solution for SDZ detection was optimized.”

  1. Line 149: using

Response: Thank you for your comment. Correct according to comments (Line 181).

  1. Figure 2A: add blanks - curves of electrolyte solutions without addition of SDZ.

Response: Thank you for your comment. We have added blanks - curves of electrolyte solutions in Figure 3 (A, e-h).

Figure 3. (A) Differential pulse diagram of sulfadiazine (SDZ) (a-d) and without SDZ (e-h) detection in different electrolyte solutions; (B) Current response of SDZ detection in different electrolyte solutions. (a, NaCl (3.8% pH 4); b, acetic acid-sodium acetate buffer solution (pH 4); c, artificial seawater (pH 4); d, phosphate buffer solution (pH 4); e-h refers to a-d electrolyte solution without SDZ).

  1. Line 156: add the preparation and origin of chemicals of all used buffer solutions to the experimental section

Response: Thank you for your comment. We have added the chemicals of all used to the section 2.1, and we also added the preparation of the buffer solutions in Line 97-105.

“Sulfadiazine (SDZ) was purchased from Aladdin Biochemical Technology Co., Ltd (Shanghai, China). Acetonitrile was purchased from Merck. Acetic acid was purchased from Tianjin Kemeiou Chemical Reagent Co., Ltd (Tianjin, China). Sodium dihydrogen phosphate (NaH2PO4), disodium hydrogen phosphate (Na2HPO4),potassium chloride (KCl), sulfuric acid (H2SO4), sodium hydroxide (NaOH), sodium chloride (NaCl), calcium chloride (CaCl2), magnesium chloride (MgCl2), sodium sulfate (Na2SO4), hydrochloric acid (HCl), sodium acetate , Strontium chloride (SrClz 6H2O), Potassium bromide (KBr), sodium bicarbonate(NaHCO3), Sodium fluoride (NaF), boric acid (H3BO3), potassium ferricyanide (K3Fe(CN)6) and ferrous iron Potassium cyanide (K4Fe(CN)6) was obtained from Sinopharm Chemical Reagent Co., Ltd (Shanghai, China). All reagents were of analytical grade, and the water was ultrapure (18 MΩ cm, 25 ℃).”

“The preparation of NaCl (3.8%) solution was to dissolve 3.8 g NaCl solid in 100 mL of water. The ABS buffer solution (pH 4) and artificial seawater was prepared as above mentioned. In addition, the pH of all used buffer solutions was adjusted with HCl and NaOH.”

  1. Line 160: investigated

Response: Thank you for your comment. Correct according to comments (Line 196).

  1. Line 164: rewrite and rethink the sentence completely. please refer to the oxidation mechanism of SDZ, these potential shifts are often a result of coupled proton/electron transfer. ionization of the analyte itself doesn´t change

Response: Thank you for your comment. We have revised the sentence as follows.

“This is because the SDZ can be electrochemically oxidized at the ‒NH2 group by two-electron and two-proton transfer process as shown in Figure S1. Besides, it reveals that the oxidation of SDZ occurs at 0.9-1.0 V, which is consistent with previous reports [29-33].”

Figure. S1. Electrochemical oxidation mechanism of sulfadiazine.

  1. Line 173: sensitivity is defined by IUPAC as the slope of calibration curve

Response: Thank you for your comment. We have modified the sentence as follows. (Line 219)

“To evaluate the analytical performance of the constructed sensor, 10 mL of SDZ standard solution was detected by DPV in ABS buffer solution at pH 4.”

  1. Line 177: use uncapitalized c

Response: Thank you for your comment. Correct according to comments (Line 223).

  1. Line 177: how was LOD calculated?

Response: Thank you for your comment. The LOD was calculated by the equations LOD = 3 σ/b (where σ is the standard deviation obtained from five times measurements of the blank signal and b is the analytical sensitivity represented by the slope of the calibration plot). We have added it in section 3.4. (Line 225-227)

“To evaluate the analytical performance of the constructed sensor, 10 mL of SDZ standard solution was detected by DPV in ABS buffer solution at pH 4. As shown in Figure 5, the concentration of SDZ and the generated peak current have a good linear relationship in the range of 20-300 μmol/L, the linear regression equation can be described as Ip (μA) = 0.0269 c (μmol/L) + 1.2, the correlation coefficient (R2) was 0.994, and the limit of detection (LOD) is 6.14 μmol/L which was calculated by the following equations: LOD = 3 σ/b (where σ is the standard deviation obtained from five times measurements of the blank signal and b is the analytical sensitivity represented by the slope of the calibration plot).”

  1. Figure 4: add blank and preferably a curve with concentration at the level of the LOD

Response: Thank you for your comment. We have added blank and preferably a curve with concentration at the level of the LOD in Figure 5.

Figure 5. (A) Differential pulse diagram of glassy carbon electrode against SDZ of different concentrations; (B) Linear relationship between different concentrations of SDZ and the oxidation peak current.

  1. Line 189: and

Response: Thank you for your comment. Correct according to comments (Line 242).

  1. Figure 6: add error bars

Response: Thank you for your comment. We have added error bars in Figure 6B.

Figure 6. Differential pulse curve (A) and current response (B) of acetic acid-sodium acetate buffer solution containing 50 μmol/L of SDZ detected by the same electrode under the same conditions (a, interfering substances added, b, without interfering substances added).

  1. Line 221: polishing!

Response: Thank you for your comment. Correct according to comments (Line 271).

  1. Line 223: after polishing

Response: Thank you for your comment. Correct according to comments (Line 273).

  1. Line 228: tool

Response: Thank you for your comment. Correct according to comments (Line 279).

We tried our best to improve the manuscript and made some changes in the manuscript. These changes will not influence the content and framework of the paper, and here we did not list the changes but marked in red in revised paper.

We greatly appreciate both your help and the helpful comments of the reviewers, and we believe a more balance and a better account of our work have been produced. We hope that the revised manuscript is acceptable for publication. Looking forward to your reply.

Thank you for your consideration!

With best regards,

Yours sincerely,

Zhengguo Cui,

[email protected]

Reviewer 2 Report

The paper describes “A low-cost electrochemical method for the determination of sulfadiazine in aquaculture wastewater.” Unfortunately, the authors did not illustrate this work's novelty clearly compared with the previously published similar work. According to my consent, the quality of the manuscript is publishable in “International Journal of Environmental Research and Public Health (IJERPH)” after addressing my comments on this manuscript.

Comments:

1.     Generally, the novelty of the manuscript is plain in the abstract and Introduction part, which cannot trigger readers' interest. The authors need to pay more attention to highlighting the innovation of current work more clearly.

2.     There are acronyms used in the manuscript without providing full form.

3.     The authors need to include the SEM of the electrode.

4.     The authors should calculate the electrochemically active surface area of electrodes.

5.     Authors need to include the electrochemical interaction mechanism.

6.     No profound explanation is provided to investigate the kinetics of the catalysts at the reaction interface.

7.     The authors must include the comparison table for SDZ detection with previous electrodes.

8.     The typos and grammatical errors are scattered throughout the paper and must be corrected with the utmost care.

Author Response

Response to Reviewer 2 Comments

Dear Prof. Aiyana Zhang and Reviewer,

We greatly acknowledge the editor and reviewer for their constructive comments and advice. Thank you for your patience and these valuable comments. It is our honor to get your help to improve our manuscript entitled “A low-cost electrochemical method for the determination of sulfadiazine in aquaculture wastewater” with the manuscript number ijerph-2058366.

We have revised the manuscript according to every single comment which made by the editor and reviewers. We check the editor and reviewers’ comments repeatedly and closely to ensure each comment is addressed. The following is our reply to the comments in red color. Moreover, the changes have been highlighted in red color in a marked copy of the revised manuscript (in submitted documents).

Response to Reviewer #2:

The paper describes “A low-cost electrochemical method for the determination of sulfadiazine in aquaculture wastewater.” Unfortunately, the authors did not illustrate this work's novelty clearly compared with the previously published similar work. According to my consent, the quality of the manuscript is publishable in “International Journal of Environmental Research and Public Health (IJERPH)” after addressing my comments on this manuscript.

Response: Thank you for your carefully reading and valuable comments. We have revised the text according to the comments. Besides, a native English speaker has revised the manuscript and we try our best to correct technical and formal flaws in the manuscript.

  1. Generally, the novelty of the manuscript is plain in the abstract and Introduction part, which cannot trigger readers' interest. The authors need to pay more attention to highlighting the innovation of current work more clearly.

Response: Thank you for your comment. We have paid more attention to highlighting the innovation of current work more clearly in the abstract and introduction of the manuscript.

The abstract was revised as follows now.

“In this paper, a new method based on the glassy carbon electrode electrochemical sensor for the detection of sulfadiazine in aquaculture wastewater was developed without using complex materials to modify the electrode surface, and detecting the sulfadiazine with electrochemically oxidizes directly, which was a simple operation and low cost. (Line 13-17)”

The introduction was revised as follows now.

“The above-mentioned electrochemical detection methods used to modify the electrode to amplify the electrochemical signal and then directly or indirectly detect SDZ in the sample. Although these methods amplified the electrochemical signal of the electrochemical sensor, they would have a higher cost for modification, complicated configuration and easy to fall off. It is simple, time-saving, and low-cost to manufacture sensors without using complex materials, and it is conducive to the implantation of this method in rapid routine analysis. In order to minimize the cost and reduce the detection time, the unmodified electrochemical sensor was characterized with differential pulse voltammetry curve and cyclic voltammetry curve. The background solution and pH of the measured SDZ were optimized. Besides, considering the complexity of the aquaculture wastewater, artificial seawater was regarded as one significant influence factor for optimization. We want to reduce the detection cost as much as possible, make the preparation and detection process of electrochemical sensors simpler. Finally, the new proposed method based on electrochemical sensor was applied to detect the SDZ in aquaculture wastewater. (Lines 79-93)”

  1. There are acronyms used in the manuscript without providing full form.

Response: Thank you for your comment. We have supplemented the complete form of acronyms in the paper.

“SDZ, sulfonamide; SAs, sulfonamide antibiotics; HPLC, liquid chromatography; GCE, glassy carbon electrode; SEM, scanning electron microscopy; CV, cyclic voltammetry; ABS, acetic acid-sodium acetate buffer solution; DPV, differential pulse voltammetry; LOD, limit of detection”

  1. The authors need to include the SEM of the electrode.

Response: Thank you for your comment. We have characterized the surface morphology of the electrode with scanning electron microscope and added it in the text as follows. (Lines 159-162)

“The surface morphology of polished and unpolished glassy carbon electrode was characterized with SEM. As shown in Figure 2, the polished electrode surface was rough which increased the electrochemical active surface area, and the unpolished electrode surface was passivated and smooth.”

Figure 2. SEM image of glassy carbon electrode. A, after polishing electrode; B, before polishing electrode.

  1. The authors should calculate the electrochemically active surface area of electrodes.

Response: Thank you for your comment. We have calculated the electrochemically active surface area of electrodes based on the Randles-Sevcik equation and added this part in the text as follows. (Lines 159-169)

“The surface morphology of polished and unpolished glassy carbon electrode was characterized with SEM. As shown in Figure 2, the polished electrode surface was rough which increased the electrochemical active surface area, and the unpolished electrode surface was passivated and smooth. In addition, the electrochemically active surface areas of GCE were calculated by the Randles-Sevcik equation (1) [22] as follows:

Ip = (2.69 × 105) n3/2AD1/2Cv1/2      (1)

where n is the number of electrons produced in the reaction, equal to 1. v is the scan rate equal to 50 mV/s, A is the electroactive surface area of the electrode, and D is the Fe(CN)63-/Fe(CN)64- diffusion coefficient equal to 7.6 × 10-5 cm2/s, C is the Fe(CN)63-/Fe(CN)64- concentration (5 mmol/L), and Ip is the peak current (A). For GCE in this study, the electroactive surface area was found to be 0.12 cm2.”

  1. Authors need to include the electrochemical interaction mechanism.

Response: Thank you for your comment. We have added the electrochemical interaction mechanism in the text as follows. (Lines 198-201)

“The SDZ can be electrochemically oxidized at the ‒NH2 group by two-electron and two-proton transfer process as shown in Figure S1. Besides, it reveals that the oxidation of SDZ occurs at 0.9-1.0 V, which is consistent with previous reports [29-33]. ”

Figure. S1. Electrochemical oxidation mechanism of sulfadiazine.

  1. No profound explanation is provided to investigate the kinetics of the catalysts at the reaction interface.

Response: Thank you for your comment. We have supplemented the dynamics parameters of electrode according to Laviron equation and added it in the paper as follows. (Lines 204-213)

“It was found that there was a linear relationship between the scanning rate and the CV curve containing SDZ electrolyte, according to which the relationship between Ep and ln v could be calculated, and the heterogeneous electron transfer rate constant (ks) could be calculated according to the Laviron equation (2) [34]:

Ep = E0 + RT/(anF) [ln [(RTks)/(anF)] - ln v]      (2)

where Ep is anode peak potential, E0 is formal potential, R is gas constant (8.314 J/mol·K), T is the temperature (298.15 K), α is the electron transfer coefficient and calculated based on the linear relationship between Ep and ln v, n is the charge transfer number, F is the Faraday constant, and Ks is the electron transfer rate constant of the surface process, υ is the scanning rate. At last, Ks was calculated as 1.13 s-1.”

  1. The authors must include the comparison table for SDZ detection with previous electrodes.

Response: Thank you for your comment. We have added the comparison table for SDZ detection with previous electrodes in the text and Supplementary Information

as follows. (Lines 228-230)

“The performance of the proposed method was compared with the previously reported SDZ sensors in the literature in Table S1. Compared with other unmodified electrodes, this sensor had satisfactory electrocatalytic performance.”

Table S1. Analytical data comparison between SDZ detection and previous electrodes.

Electrode

Electrolyte solution

LOD

Linear range

Ref.

GCE

0.04 M B-R buffer

(pH 6.8)

10.9 μM

62.7-340 μM

[1]

Carboxyl-MWCNTs/GCE

0.04 M B-R buffer 

(pH 2)

0.07μM

0.50-110 μM

[2]

MIP/GO@COF/GCE

0.2 M PBS

(pH 7)

0.16 μM

0.5–200 μM

[3]

MWCNT-GCE

0.04 M B-R buffer

(pH 7)

7.1 μM

10-2000 μM

[4]

GCE

0.1M ABS

(pH 4)

6.14 μM

20-300 μM

This work

(GCE, glassy carbon electrode; MWCNT, multiwalled carbon nanotube; MIP, molecularly imprinted polymer; GO@COF, grapheneoxide@covalentorganic framework; B-R buffer, Britton–Robinson buffer; PBS, phosphate buffer; ABS, acetic acid-sodium acetate buffer solution).

  1. The typos and grammatical errors are scattered throughout the paper and must be corrected with the utmost care.

Response: Thank you for your comment. We have carefully checked and corrected the spelling and grammar errors in the article. In addition, a native English speaker has revised the manuscript and we made an examination and revision of the article.

We tried our best to improve the manuscript and made some changes in the manuscript. These changes will not influence the content and framework of the paper, and here we did not list the changes but marked in red in revised paper.

We greatly appreciate both your help and the helpful comments of the reviewers, and we believe a more balance and a better account of our work have been produced. We hope that the revised manuscript is acceptable for publication. Looking forward to your reply.

Thank you for your consideration!

With best regards,

Yours sincerely,

Zhengguo Cui,

[email protected]

Round 2

Reviewer 2 Report

 The authors answered all questions and completed the changes. The manuscript in its current form is acceptable.